# Size-Controllable Nanosystem with Double Responsive for Deep Photodynamic Therapy

**DOI:** 10.3390/pharmaceutics15030940

**Published:** 2023-03-14

**Authors:** Shuang-Shuang Wan, Jun Tao, Qian Wu, Wu-Rui Liu, Xian-Guang Ding, Xian-Zheng Zhang

**Affiliations:** 1State Key Laboratory of Organic Electronics and Information Displays, Jiangsu Key Laboratory for Biosensors, Institute of Advanced Materials (IAM), Nanjing University of Posts and Telecommunications, 9 Wenyuan Road, Nanjing 210023, China; 2Key Laboratory of Biomedical Polymers of Ministry of Education, Department of Chemistry, Wuhan University, Wuhan 430072, China

**Keywords:** cancer, photodynamic therapy, upconverting nanoparticle, self-assembly, metal-organic framework

## Abstract

Photodynamic therapy (PDT) is a promising strategy for cancer treatment. However, a poor tissue penetration of activation light and low target specificity seriously hindered the clinical application of PDT. Here, we designed and constructed a size-controllable nanosystem (UPH) with inside-out responsive for deep PDT with enhanced biosafety. To obtain nanoparticles with the best quantum yield, a series of core-shell nanoparticles (UCNP@nPCN) with different thicknesses were synthesized by a layer-by-layer self-assembly method to incorporate a porphyritic porous coordination network (PCN) onto the surface of upconverting nanoparticles (UCNPs), followed by coating with hyaluronic acid (HA) on the surface of nanoparticles with optimized thickness to form the UPH nanoparticles. With the aid of HA, the UPH nanoparticles were capable of preferentially enriching in tumor sites and specific endocytosis by CD44 receptors as well as responsive degradation by hyaluronidase in cancer cells after intravenous administration. Subsequently, after being activated by strong penetrating 980 nm near-infrared light (NIR), the UPH nanoparticles efficiently converted oxygen into strongly oxidizing reactive oxygen species based on the fluorescence resonance energy transfer (FRET) effect, thereby significantly inhibiting tumor growth. Experimental results in vitro and in vivo indicated that such dual-responsive nanoparticles successfully realize the photodynamic therapy of deep-seated cancer with negligible side effects, which showed great potential for potential clinical translational research.

## 1. Introduction

Photodynamic therapy is a promising treatment modality that requires three elements including photosensitizers, oxygen, and excitation light. The process of PDT is that the photosensitizer absorbs specific wavelengths of light to be activated to the excited state, which in the process of returning to ground state transfers energy to oxygen to generate reactive oxygen species that would cause damage to cells. [1,2,3]. PDT has attracted widespread research attention because of its spatiotemporal controllability, minimal invasiveness, and low side effects [4,5]. However, the optimal excitation wavelength of commonly used photosensitizers is in the UV-vis windows (200–700 nm) [6]. This short wavelength has a poor tissue penetration [7], thus seriously hindering the clinical application of PDT in the treatment of large or internal tumors. Moreover, due to the lack of targeted and unavoidable daylight activation [8], photodynamic therapy can lead to low therapeutic efficiency [9] and increase the pain and burden of patients [10].

In order to address the issue of light penetration, many near-infrared (NIR) light excitation strategies [11,12,13] for deep-tissue PDT have been proposed, especially upconversion luminescence [14,15]. Different from traditional energy down-conversion based fluorescence luminescence [16], upconversion luminescence is a process capable of converting two or more low-energy photons that have been absorbed into the emission of photons with a relatively high energy [17], which dramatically reduces the autofluorescence background and improves tissue penetration as well as diminishes phototoxicity [18]. As an important upconversion material, the lanthanide upconversion nanoparticles were constructed by inorganic substrates such as oxides, fluorides, and halogen oxides by doping with trivalent rare-earth ions (such as Er^3+^, Eu^3+^, Yb^3+^, Tm^3+^, Ho^3+^, etc.), which were endowed with the ability to convert NIR light into UV-vis light that perfectly matches the absorption spectrum of most photosensitizers for efficient activation of photosensitizers [19]. In view of the super superior optical properties including high photostability, large anti-Stokes shift, sharp multi-wavelength emission bandwidth, weak autofluorescence, and deep tissue penetration [20], UCNPs as ideal nanotransducers are widely used NIR light-medicated PDT for deep-seated tumors. Additionally, to further improve therapeutic efficiency, many efforts should be devoted to developing smart UCNP-based PDT theranostic agents for preferential enrichment and intelligent responsiveness at targeted site. 

Hyaluronic acid (HA) is a naturally occurring, biodegradable, biocompatible, and non-immunogenic linear polysaccharide present in the extracellular matrix and synovial fluid [21], showing great promise in the field of biomedical application [22,23]. Of special note, HA have been documented to bind CD44 receptors highly overexpressed on tumor cells, making it a potential active targeting ligand [24,25]. In addition, HA can be rapidly degraded by hyaluronidases abundant in cytoplasm of tumor cells [26], which trigger enzyme-responsive drug release at the tumor site [27]. These properties of HA make it a popular targeted delivery for tumor treatment. 

In this work, we designed a size-controlled UCNP-based theranostic agent for deep PDT with enhanced safety. As depicted in Figure 1, we first designed and synthesized the core-shell nanosystem (UCNP@nPCN) with different thicknesses, and then further coated the optimized nanoparticles (UCNP@10PCN) with HA to form the UPH nanoparticles. After intravenous administration, the UPH nanoparticles were expected to preferentially accumulate at tumor sites, be subsequently endocytosed by tumor cells, and trigger enzyme-responsive degradation. Ultimately, the UPH nanoparticles evoked PDT in response to NIR light, successfully suppressing tumors growth in mice.

## 2. Materials and Methods

### 2.1. Materials

Yttrium oxide (Y_2_O_3_), Ytterbium oxide (Yb_2_O_3_), Erbium oxide (Er_2_O_3_), and Polyacrylic acid polymers (PAA) were taken by Aladdin-Reagent Co. Ltd. (Shanghai, China). Zirconyl chloride octahydrate (ZrOCl_2_·8H_2_O) and benzoic acid (BA) were purchased from Sinopharm Chemical Reagent Co., Ltd., (Shanghai, China). Tetrakis (4-carboxyphenyl) porphyrin (TCPP) was synthesized based on the previous procedure with some modifications [1]. 2′,7′-dichlorofluorescin diacetate (DCFH-DA), Annexin V-FITC/PI Cell Apoptosis Kit, and Calcein were obtained from Beyotime Institute of Biotechnology (Shanghai, China) ScalaSansLF-Regular. All cell lines (SCC-7 and COS-7) were purchased from China Centre for Type Culture Collection (CCTCC). TEM images were carried out using Tecnai G20 S-TWIN operated at 200 kV. SEM images were taken by a field emission scanning electron microscope (Sigma). PXRD analysis was tested by Rigaku MiniFlex 600 X-ray diffractometer with Cu (K_α_ = 1.5418 Å). The hydrodynamic size and zeta potential were performed by dynamic light scattering (DLS) on a Malvern Zetasizer Nano-ZS ZEN3600 instrument. 

### 2.2. Methods

#### 2.2.1. Synthesis of UCNPs

Thermal decomposition was used to synthesize high-quality UCNP nanocrystals. In general, 15 g rare earth oxides (Y_2_O_3_, Yb_2_O_3_, Er_2_O_3_) were added to 10% water in trifluoroaceticacid and stirred at 80 °C overnight. Then the reaction mixture was concentrated under vacuum and dried for using. Subsequently, 20 mmol Re(CF_3_COO)_3_ at the molar ratio of 78%:20%:2% (Y:Yb:Er) was added in octadecene with 50% oleic acid and stirred at 130 °C under vacuum for 1 h to remove the residual water. A total of 10 mL of the above stock solution was cooled and stirred at 50 °C for another 1 h, followed by heating-up to 120 °C until no bubbles appeared. Thereafter, 0.2 g NaOH and 0.3 g NH_4_F were dissolved in 8 mL methanol and added into the mixture. After reaction at 100 °C under vacuum for 30 min, the mixture was rapidly heated to 310 °C and kept at this temperature for 30 min in a nitrogen atmosphere. Finally, the UCNP nanocrystals were obtained by precipitation with ethanol and centrifugation. 

#### 2.2.2. Synthesis of PAAylated UCNPs

PAA-UCNPs were prepared based on the method of ligand exchange of OA-stabilized UCNP with PAA. Typically, 4 mL of OA-UCNPs (50 mg/mL) was dispersed in ethanol solution (20 mL) with 0.5 M hydrochloric acid. After stirring at room temperature for 4 h, the mixture was washed with water repeatedly. Then, the obtained nanoparticles were added in a solution of PAA (50 mg/mL, pH 7.0) and stirred gently for 24 h. Finally, as-synthesized PAA-capped UCNPs were collected by centrifugation to remove excess PAA and freeze-dried for further use. 

#### 2.2.3. Preparation of UCNP@nPCN Nanocrystals

The PCN shell were coated by a layer-by-layer growing method. PAA-modified UCNPs were dispersed in 5 mL DMF and then 10 mg ZrOCl_2_·8H_2_O was added. The obtained mixture was stirred gently in a sealed vial at 90 °C for 30 min. After cooling to room temperature, the solution was subjected to washing with DMF by centrifugation at 10,000 rpm for 10 min. Subsequently, 5 mg/mL of the ligand TCPP was added to coordinate with ZrO^2+^ on the surface of PAA@UCNP. After that, a thin layer of PCN formed. Repeatedly adding ZrOCl_2_·8H_2_O and TCPP ligand for five times as described above, we obtained nanomaterials with five layers of PCN (UCNP@5PCN). Based on the same approach, we can synthesize UCNP@10PCN and UCNP@15PCN nanoparticles with different thicknesses by altering the layer number of PCN. The crude products UCNP@nPCN were purified by washing three times in DMF (10,000 rpm, 10 min), and the final products were saved in DMF for further use.

#### 2.2.4. Synthesis of the UPH Nanoparticles

In all, 4 mg HA and 2 mg UCNP@10PCN were mixed in 10 mL water and stirred overnight. The crude product was collected and purified by centrifugation with water. 

#### 2.2.5. ROS Generation Test

p-nitrosodimethy laniline (RNO) was chosen as the probe to detect the ROS. In a typical experiment, the samples (an equivalent UCNP amount of 50 μg/mL), RNO (50 μL, 25 μM), and histidine (100 μL, 9 μM) were mixed in 1 mL water. The UV absorbance of RNO at 440 nm were recorded at preset times after 980 nm light irradiation (100 mW/cm^2^). The ROS generation ability was defined as A_t_/A_0_, where A_0_ refers to initial absorbance wavelength. 

#### 2.2.6. Cellular Uptake

The endocytosis of samples in vitro by SCC-7 and COS-7 cells were evaluated via CLSM (Nikon C1-si TE2000, Japan) and flow cytometry (BD FACSAria™ III, USA). The two kinds of cells were, respectively, seeded in 6-well plates and incubated in 1 mL medium for 24 h. Then, the medium was replaced with various materials (UPH, UP and UPH nanoparticles) for 4 h incubation. Then the culture medium was removed, followed by repeated wash with PBS. Finally, the cells were observed under CLSM and collected for flow cytometry analysis. 

#### 2.2.7. Intracellular Reactive Oxygen Detection

Qualitative study of the generation of the ROS was carried out using fluorescence spectrum based on emission wavelength of DCFH at 525 nm. SCC-7 cells were seeded in glass-bottomed dishes incubated in a humidified atmosphere containing 5% (*v*/*v*) CO_2_ at 37 °C for 24 h. Thereafter, the medium was removed and the samples in the fresh one were added. Four hours later, the cells were washed with PBS and incubated with 1 mL DCFH-DA (10 μmol/L) for another 30 min. Before observation by CLSM, SCC-7 cells were washed and subjected to irradiation under the 980 nm (100 mW/cm^2^) laser for 30 s. 

#### 2.2.8. Cytotoxicity Assay

MTT assay was conducted to evaluate the toxicity against SCC-7 cells. Briefly, cells were seeded on 96-well plates at 10^4^/well and incubated for 24 h. Then, the original medium was replaced with the fresh one containing the samples. After 4 h post incubation, the cells received 3 min light irradiation (980 nm, 100 mW/cm^2^) and were cultured for an additional 24 h. Thereafter, 3-(4,5)-dimethylthiahiazo (-z-y1)-3,5-di-phenytetrazoliumromide (MTT, 20 μL, 5 mg/mL) was added for 4 h incubation, followed by elimination of the supernatant and addition of 150 μL DMSO. The cytotoxicity was defined as follows: cell viability (%) = OD_(sample)_/OD_(control)_ × 100%, in which OD_(sample)_ and OD_(control)_ refer to the value of the optical density at 570 nm in the presence of the sample or not, respectively. 

#### 2.2.9. Live–Dead Cell Staining Assay

SCC-7 cells were seeded on 6-well plates and incubated for 24 h. In all, 100 μL of various samples was added for 4 h incubation. Then, cells were irradiated with 980 nm laser (100 mW/cm^2^) for 3 min. After 24 h, the cells were stained with Calcein-AM (1 μL/well, 4 × 10^−6^ M) and PI (10 μL/well) for 15 min. Finally, cells were washed with PBS and observed by fluorescence inverted microscope.

#### 2.2.10. In Vivo Optical Imaging

All live animal experiments were performed in compliance with the guidelines of the Institutional Animal Care and Use Committee of the Animal Experiment Center of Wuhan University (Wuhan, China, approval code: ZN2021232). Live tumor model was constructed by subcutaneous injection of SCC-7 cells on the back of the BALB/c-nu mice (~20 g). Mice were intravenously injected with samples (TCPP dosages: 5 mg/kg) when the volume of tumor reached 200 mm^3^. At the preset time, mice were anesthetized and imaged by IVIS small animal imaging system (PerkinElmer). 

#### 2.2.11. Evaluation of Antitumor Effect

Once the tumors reached an approximate size of 100 mm^3^, SCC-7 tumor-bearing mice were randomized to six groups. Then, the mice were intravenously injected with various samples (150 µL), including PBS buffer, UP, and UPH. One of the two groups with PBS treatment was subjected to irradiation with a 980 nm laser at a power of 300 mW/cm^2^ for 10 min after 24 h, as were the mice injected with UP and UPH. The tumor volume and weight were recorded every day. The tumor volume was calculated based on the formula: a × b^2^/2, in which a and b refers to the length and width of tumor, respectively. After 14 days, the mice were sacrificed and anatomized to obtain the organs and tumor for histological analysis.

#### 2.2.12. Statistical Analysis

The statistical analysis was performed by two-tailed Student’s t-tests using Microsoft Excel 2013. Statistical difference was considered significant at a value of *p* < 0.05.

## 3. Results and Discussion

### 3.1. Controllable Synthesis and Characterization of UCNP@nPCN

UCNP@nPCN were prepared by layer-by-layer self-assembly. The ligands (Zr and TCPP) of the PCN were conjugated to the surface of the UCNP nanocrystal through electrostatic and coordination, and different thicknesses of the UCNP@nPCN nanomaterials were synthesized by controlling the number of coating layers. As shown in Figure 1A and Appendix A, the images of scanning electron microscopy (SEM) and transmission electron microscopy (TEM) indicated the successful synthesis of the hexahedral UCNP and the PCN nanoparticles with good dispersion and uniform size. After conjugating with the PCN, the core-shell structure of the UCNP@nPCN nanoparticles could be clearly observed, and as the number of layers increased, the thickness of the shell also increased significantly. When the number of layers was 5, the thickness of shell was close to 5 nm, and it would reach 20 nm after coating 15 layers of the PCN. Using this correlation between the number of layers and the thickness, we could customize the nanoparticles with the thickness as required. More importantly, the nanoparticles synthesized by this method were very uniform and monodisperse, which was beneficial for biomedical applications. In addition, the size (Figure 1B) and potential (Appendix A) also indicate that the core-shell UCNP@nPCN nanoparticles were successfully synthesized. Consistent with the TEM images, the size of UCNP@nPCN nanoparticles also reflected the thickness of the PCN coating. Further, to determine whether the self-assembled method synthesized the PCN, we examined the crystal structure of the nanoparticles by powder X-ray diffraction. The characteristic crystal peaks of the PCN can be clearly observed in the UCNP@15PCN nanoparticles when 15 layers are coated. The indistinct peaks in the UCNP@5PCN and UCNP@10PCN nanoparticles might be attributable to the insufficient crystallinity caused by the low thickness. 

With the knowledge of key role of the ROS in PDT, the ROS production ability of the UCNP@nPCN nanoparticles was conducted after successful synthesis by LBL assembly. Prior to the measurement, UV-visible absorption and fluorescence spectrum were tested to ensure the energy transfer from the UCNP donor to the PCN acceptor via fluorescence resonance energy transfer (FRET) for photodynamic therapy. In Figure 1C, all absorption peaks of UCNP@nPCN appeared in the Soret band and four Q-bands, coinciding with the peaks of the PCN nanoparticles. As the thickness increased, the absorption of the UCNP@nPCN nanoparticles increased, which was beneficial for the ROS generation. A slight red shift might be attributable to the ubiquitous scattering effect of the nanomaterials [28]. Then, the spectral overlap between the PCN absorption and the UCNP emission (Figure 1D) illustrated the possibility of FRET between the UCNP and PCN. More importantly, fluorescence emission of UCNP@nPCN was quenched significantly with the increase thickness of the PCN shell (Appendix A), illustrating excellent FRET ability. Under this premise, we further explored the ability of the UCNP@nPCN nanoparticles to produce singlet oxygen. It had been reported that ^1^O_2_ was capable of quenching the characteristic absorption wavelength of RNO at 440 nm [29]. As a singlet oxygen probe, the RNO absorption (Figure 1E) in the UCNP@nPCN group decreased with 980 nm laser exposure time. However, there was almost no singlet oxygen in the single component (UCNP and PCN group). These phenomena demonstrated successful FRET for efficient singlet oxygen generation. Notably, among all the core-shell nanoparticles, UCNP@10PCN had the best effect on ^1^O_2_ generation, nearly 67% of the reduced RNO. Therefore, the UCNP@10PCN nanoparticles were selected as the optimal material for subsequent antitumor research.

### 3.2. HA-Targeted Enhanced Endocytosis

Considering the protective function of HA to avoid metabolic clearance in long blood circulation, the strong affinity to the CD44 receptors overexpressed on the surface of many tumor cells, and the easy degradation by hyaluronidase (HAase) in the cell to release payload [30], HA was coated on the surface of UCNP@10PCN to serve as an active targeting ligand for the enhanced endocytosis of the nanoparticles. By using coordination and electrostatic interactions, we successfully conjugated HA on the surface of UCNP@10PCN to form the UCNP@10PCN@HA (UPH) nanoparticles, which was confirmed by dynamic light scattering and infrared spectroscopy (Appendix A). In addition, the UPH nanoparticles showed a good stability in the PBS buffer (Appendix A). To verify the enzyme responsiveness of the UPH nanoparticles, we co-incubated the UPH material with hyaluronidase. It was clear that the degradation of enzymes accelerated the production of the ROS (Appendix A). This may be attributed to the consumption of the ROS by hyaluronidase. Subsequently, we conducted an endocytosis experiment against tumor cells (SCC-7, known for highly abundant CD44 receptors) and normal cells (COS-7). As shown in Figure 2A, a larger area of red fluorescence was observed in the cytoplasm of the SCC-7 cells compared with that of the COS-7 cells, demonstrating the higher uptake of the UPH nanoparticles by tumor cells. To further demonstrate HA-mediated endocytosis, the mixture of HA and UPH was co-incubated with cells. Due to the competition mechanism in the presence of HA, the uptake of the UPH nanoparticles was drastically decreased. The phenomenon indicated that the high affinity of HA for cancer cells contributed to the endocytosis of the nanoparticles. Furthermore, the cellular uptake of the UPH nanoparticles was quantified using flow cytometry. In Figure 2B,C, UPH had the greatest uptake in the SCC-7 cells in all groups, nearly 23 times that of the UPH+HA group in the COS7 cells.

### 3.3. Intracellular ROS Generation and Cytotoxicity Assessment

Motivated by the good endocytosis results, we then used CLSM to visualize the intracellular ROS production in vitro after irradiation with 980 nm laser for 2 min. Dichlorofluoresceindiacetate (DCFH-DA) was the chosen ROS probe, which could be oxidized to 2′,7′-dichlorofluorescin (DCF) by the ROS with green fluorescence. After the SCC-7 cells were incubated with various samples and stained with DCFH-DA, intracellular green fluorescence was observed with CLSM. As shown in Figure 3A and Appendix A, almost no fluorescent in the tumor cells treated with UPH in the absence of light irradiation, indicating the UPH nanoparticles-medicated ROS generation, required the activation of the external 980 nm laser. Moreover, a negligible fluorescent signal could be detected in UCNP + hv and PCN + hv groups, similar to the results in the solutions (Figure 1E). This demonstrated that FRET was a necessary condition for the system to generate the ROS. Of special note, attributed to high cellular uptake of the UPH nanoparticles, the UPH + hv group presented bright green fluorescence compared with the UP + hv, indicating sufficient ROS generation for potential FRET-base PDT. 

Next, we studied the killing effect of the UPH nanomaterial on tumor cells in vitro. As shown in Figure 3B, the control samples (UCNP and PCN) had little toxicity to cancer cells regardless of irradiation or not even at high doses, due to almost no ROS production. Moreover, since there was no external laser to activate the photodynamic therapy, the cell viability of the SCC-7 cells was high in the UPH and UP group, indicating the good biocompatibility of the samples. In contrast, cells treated with UP under 980nm laser showed higher death rates. This tumor-killing effect was further enhanced by coating with HA. We could find a concentration-dependent cytotoxicity in the UPH + hv group, only 37.9% of cell viability at the highest concentration. Such an excellent antitumor effect of the UPH + hv treatment was confirmed by live–dead cell staining experiments. In Figure 3C, in agreement with the cytotoxicity data, the ratio of live to dead cells (green/red fluorescence) was highest in the presence of 980 nm radiation for the cells treated with UPH in all treatment groups.

### 3.4. Tumor-Targeted Imaging and Deep Antitumor Effect In Vivo

To demonstrate tumor targeting of the UPH nanoparticles, tumor-bearing mice were intravenously administrated with samples and imaged at preset times by a small animal imaging system. As shown in Figure 4A, the fluorescence signal of the UPH nanoparticles presented a trend of increasing and then decreasing with time. That meant that HA made the UPH nanomaterials be preferentially enriched in the tumor site and reach the peak at 24 h, followed by gradual removal due to metabolism. Compared with the UP group, the UPH group with strong fluorescence has more advantages in treatment by prolonging the retention time at the tumor site and reducing metabolic clearance as well as decreasing systemic side effects, which was also confirmed by the distribution of nanomaterials in organs. 

Furthermore, the excellent imaging performance of the UPH nanomaterials motivated us to explore antitumor efficacy of UPH under NIR light irradiation. When the tumor volume of the mice reached 100 cm^3^, the mice were subjected with treatment by intravenously injecting the nanomaterials. A slight change of mice body weight (Figure 4B) was observed throughout treatment, illustrating the negligible systemic toxicity of the drug formulation. The same conclusion was proved by the H&E slice of the main organs (Appendix A). As shown in Figure 4C, through the comparison of the PBS groups with irradiation or not, no difference in tumor suppression in the two groups indicated that 10 min of 980 nm laser irradiation had little effect on the mice tumors. Since the generation of the ROS depended on lasers, the pure nanomaterials could not inhibit the tumor without laser excitation. In sharp contrast, the UP + hv group was almost twice as effective at suppressing tumors as the control group PBS group, implying that the nanoparticles accumulated at tumor sites via the enhanced permeability and retention (EPR) effect enabled induce tumor cell death after the NIR laser activation. Surprisingly, after being coated with HA, the ability of the UPH nanomaterials to suppress tumors was further improved, and the tumors were almost invisible after three treatments. This should be attributed to the high enrichment and long retention mediated by the active targeting of HA demonstrated above. On the 15th day of treatment, all mice were anesthetized to obtain tumor tissue and major organs for histological analysis. Clearly, consistent with the tumor volume results, the UP + hv group had the smallest ex vivo tumors (Figure 4D), showing the best treatment effect. Furthermore, the tumor tissue treated with UP + hv exhibited the fewest tumor cells (purple fluorescent) in the images of the HE section (Figure 4E). Therefore, in vivo experiments confirmed that the UPH nanomaterial could achieve deep photodynamic therapy-mediated antitumor effects through FRET effects. In contrast to conventional photodynamic therapy, we had addressed the issue of light penetration using the upconversion nanoparticles and demonstrated that this strategy enables deep cancer treatment. On the basis, the UNCP@10PCN nanoparticles were further coated with hyaluronic acid, whose targeting and degradability further enhanced the treatment efficacy. This work offered a compelling strategy by coating the nanomaterials with hyaluronic acid for reducing synthesis complexity of the nanomaterials while retaining several natural benefits such as active targeting and stimuli-responsive degradation.

## 4. Conclusions

In summary, we successfully constructed the UCNP-based theranostic agents with controllable size and dual response for PDT of large and internal tumors. By the layer-by-layer self-assembly approach, we conjugated the PCN nanoparticles to the surface of the UCNP, thereby synthesizing a series of nanoparticles with different thicknesses by controlling the number of the PCN outer layer. Experimental results suggested that the 10-layered nanoparticles (UCNP@10PCN) have the best imaging effect and ROS quantum yield. In view of the excellent biological properties, HA was chosen as a coating to encapsulate the UCNP@10PCN nanoparticles, which, in order to further enhance tumor targeting and avoid side effects, were coated with HA, which endows the material with a high capacity for tumor targeting, cellular uptake, and responsive degradation. Ultimately, in response to external NIR light and internal enzymatic stimulation, the UPH nanoparticles could effectively scavenge tumor cells via FRET-medicated highly toxic ROS generation. Live animal models also demonstrated that such a dual-responsive UPH nanomaterial could thoroughly suppress tumor growth accompanied with lower side effects by deep PDT, which paved the way for clinical translation of deep PDT.

## Data Availability

The data that support the findings of this study are available from the corresponding author upon reasonable request.

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
