# Peer review of "Size-Controllable Nanosystem with Double Responsive for Deep Photodynamic Therapy"

_pharmaceutics, 2023, doi:10.3390/pharmaceutics15030940_

Round 1

Reviewer 1 Report

The manuscript is devoted  a current and fascinating topic, design and construction a size-controllable nanosystem with inside-out responsive for deep PDT with enhanced biosafety, which is clearly and concisely presented. The original results are obtained, which could be useful for the forthcoming experimental investigations. The text is well organized and quite well written, but small changes are necessary:

1) Despite the title "results and discussion" section, very little discussion of presented in vivo data is provided. The discussion should be added, in which you can discuss the advantages of your model and compare your results with the latest reports, in particular, devoted hyaluronic acid-based nano drug delivery systems.

2) The section on materials and methods lacks a description of the statistical methods of research.

3) The scale bar is missing from some of the photos, such as Figure 3 and Figure S9.

In conclusion – the manuscript could be accepted for publication after minor changes

Reviewer 2 Report

1. In the title, authors emphasizes size-controllable nanosystem. The stability f particle size distribution of UCNP@nPCN should be presented according time-course.

2. The differences between UCNP@5PCN, UCNP@10PCN and UCNP@15PCN should be summarized in the separated Table.

3. What is the reason for higher RNO concentration of UCNP@10PCN compared to UCNP@15 PCN ?

4. Chemical structure UCNP@nPCN and UPH should be indicated in the separated Figure. Especially, FT-IR is not empugh (Figure S6) to confirm chemical structure of your nanosystem. NMR is recommended.

Reviewer 3 Report

This manuscript describes the development of a nanosystem for improved and safer photodynamic therapy (PDT) of cancer. The strategy was to design size-controlled lanthanide up conversion nanoparticles (UCNP) as theranostic agent for deep PDT with improved safety. Firstly, the authors designed and synthesized core-shell nanosystem (UCNP@nPCN) composed of UCNP and porphyritic porous coordination network (PCN) with different thicknesses, Furthermore, they coated the optimized nanoparticles (UCNP@10PCN) with hyaluronic acid (HA) to obtain targeted and size-controllable nanoparticles (UPH). The manuscript’s subject is very interesting. However, the manuscript shows some points to be revised:

- Title must be revised to describe the work more properly.

- There are many acronyms in the text and the authors must describe them at the first time in introduction, materials or in methods.

- Page 1 - Introduction - First paragraph - PDT is a modality that uses a sensitizer only to produce ROS? Explain better the PDT fundamentals.

- Page 2 - line 4 - Revise “improves”.

- Overall, the introduction must be improved showing better the PDT mechanism, the fundamentals and applications of lanthanide up conversion nanoparticles and PCN, to better understanding of UPH nanoparticles.

- Page 2 - Materials - This section should describe only the materials. Equipments must be described in each methodology.

- Page 2 - Methods - What does “Re(CF3COO)3 mean and what is the function?

-  Page 4 - For in vitro biological studies, the authors must inform the origin of type of cells like SCC-7 and COS-7.

- Page 4 - 2.2.6. - What do DCFH and DCFH-DA mean?

- Page 5 - In vivo studies - The authors must inform the number of document from the Institutional Animal Care and Use Committee of the Animal Experiment Center that approved the performing the experiment.

- Figure 1 - The figure caption must be self-explanatory. So, it must be improved with more information, mainly about the acronyms.

- Figure 2 - The same as figure 1.

- Figure 3 - The subfigure B must be better explained in the caption.

- Figure 4 - The subfigure D must be better explained din the caption. What do the 4 deferent columns of tumors? Are they 4 replicates? Subfigure D must be better explained using arrows.

- Supplementary material - The captions of figures S1, S2, S3, S5, S6 and S8 must be improved to better explain the details in each one.

- Supp. material - Figure S9 - The caption should be revised to describe better the alterations and informations. The authors must use arrows to show the events. Size bars and magnification are also important to include.

Reviewer 4 Report

The manuscript titled “ Size controllable nanosystem with inside out responsive for FRET-based deep photodynamic therapy,” presents the application of upconversion nanoparticles (UCNPs) with surface modification for deep-PDT. This work is directed to resolve the issues pertaining to the inherent limitation of low depth penetration in conventional PDT, and also to incorporate specificity for achieving targeted delivery of the drug thereby reducing the off-target induced harmful effects. However, conceptually the idea is well thought, but lacks proper execution, which can be evidently seen throughout the manuscript. The manuscript in its current form fails to scientifically prove the concept and possesses a lot of gaps to reach a justified conclusion. In addition, there is inconsistency regarding the selected formulation and the formulation that has been characterized. To be more specific, the authors claim to have selected UCNP@6PCN in one section and UCNP@10PCN in another section (see in major comments) but they failed to show any characterization of the selected 6-layer NPs, and instead characterized 5, 10 and 15-layer analogues. There are some major revisions and some experimental gaps that needs to be addressed before the manuscript can be reconsidered.

Major:

-       The title claims that there exists FRET phenomena, but it is not well-supported within the manuscript, as there lacks any evidence showing the occurrence of FRET. Fluorescence spectra as a stand-alone study does not prove the presence of FRET. Further investigated would be required to validate and confirm the claim around FRET.

-       The authors depict in the scheme that HA coating undergoes enzymatic degradation once inside the cytoplasm. What is the relevance of this modality in the current work? If it is a future possibility and application, then why to include in present work since this is not being tested.

-       Experimental sections lack details that are imperative for reproducibility of the results. The method section lacks details of the samples tested. It will be critical to identify the samples and the concentrations tested.

o   For example:

§  Page 4 (Section 2.2.3.) – Author mentions “thickness can be controlled be altering the layers number”,but fails to provide the experimental details for controlling the number of layers being deposited. How can the layers number be controlled? Author must include the relevant details or include a reference for the reader. In addition, procedure lacks steps taken for purifying, collecting and storing the synthesized material.

§  Page 4 (Section 2.2.4.) – Author mentioned “UCNP@6PCN, whereas in the introduction section Page 2, it is claimed that UCNP@10PCN was used for HA modification. Please clarify the discrepancy.

§  Page 4 (Section 2.2.4.) – Light irradiation conditions are missing, such as fluence rate/ dose/irradiation time etc.

§  Page 4 (Section 2.2.5.) – Experimental details left out for FACs analysis, such as sample preparation etc. Author fails to explain the samples studied for cell uptake. Please include the details for “various materials” tested.

§  Page 4 (Section 2.2.6.) – The concentration of DCFH-DA used to measure the ROS. If manufacturer’s protocol was followed, please specify.

-       TEM/DLS/SEM/Zeta potential for PAA-UCNP is missing. Since this is the formulation used to synthesize the PCN self-assembled layer, it is an important control for the UCNP@nPCN materials.

-       Page 6 : the authors claim that size and zeta potential measurements validate core-shell NPs is incorrect, especially without having the correct controls, such as mentioned in previous point.

-       There is no clear evidence to confirm the formation of UCNP@nPCN. Elemental analysis and mapping may help elucidate and support the claim.

-       Absorption spectra for UCNP is missing.

-       The authors claim of UCNPs fluorescence quenching as a function of increasing PCN thickness is partially valid, as quenching of donar should lead to fluorescence of acceptor, which is not shown to validate FRET.

-       For ROS in solution studies, (Irradiation + singlet oxygen probe) control is missing.  

-       Page 6: The authors claimed UCNP@6PCN had the best performance but failed to present and show the results for the same.

-       For DCFH-DA studies, (hv) only and (DCFH-DA) only control is missing.

-       Same for Live-Dead cell assay and viability assay.

Round 2

Reviewer 3 Report

-

Author Response

Thank you for your advice.

Reviewer 4 Report

From the Author's response letter :

Refering to Point 1: 

The authors modified the title, however there is remains lack of evidence to support "dual responsive"ness of the formulation. It will be important to explain the "duality" in the formulation.  

Refering to Point 2 : 

The authors hypothesize that HA layer will alter the diffusion of ROS (singlet oxygen), however there is no supporting experimental evidence (showing difference in ROS detected with and without HA)  nor a reference that can affirm to the hypothesis and thus there is still lack of clarity on the role of HA "degradation" in improving the efficacy of the formulation. 
